# Microtubule Cytoskeletal Network Alterations in a Transgenic Model of Tuberous Sclerosis Complex: Relevance to Autism Spectrum Disorders

**DOI:** 10.3390/ijms24087303

**Published:** 2023-04-15

**Authors:** Magdalena Gąssowska-Dobrowolska, Grzegorz A. Czapski, Magdalena Cieślik, Karolina Zajdel, Małgorzata Frontczak-Baniewicz, Lidia Babiec, Agata Adamczyk

**Affiliations:** 1Department of Cellular Signalling, Mossakowski Medical Research Institute, Polish Academy of Sciences, Pawińskiego 5, 02-106 Warsaw, Poland; 2Electron Microscopy Research Unit, Mossakowski Medical Research Institute, Polish Academy of Sciences, Pawińskiego 5, 02-106 Warsaw, Poland

**Keywords:** tuberous sclerosis complex, MAP-Tau, MAPs, cytoskeleton dysfunction, autism, animal models, TEM

## Abstract

Tuberous sclerosis complex (TSC) is a rare genetic multisystem disorder caused by loss-of-function mutations in the tumour suppressors *TSC1*/*TSC2*, both of which are negative regulators of the mammalian target of rapamycin (mTOR) kinase. Importantly, mTOR hyperactivity seems to be linked with the pathobiology of autism spectrum disorders (ASD). Recent studies suggest the potential involvement of microtubule (MT) network dysfunction in the neuropathology of “mTORopathies”, including ASD. Cytoskeletal reorganization could be responsible for neuroplasticity disturbances in ASD individuals. Thus, the aim of this work was to study the effect of Tsc2 haploinsufficiency on the cytoskeletal pathology and disturbances in the proteostasis of the key cytoskeletal proteins in the brain of a TSC mouse model of ASD. Western-blot analysis indicated significant brain-structure-dependent abnormalities in the microtubule-associated protein Tau (MAP-Tau), and reduced MAP1B and neurofilament light (NF-L) protein level in 2-month-old male B6;129S4-Tsc2^tm1Djk^/J mice. Alongside, pathological irregularities in the ultrastructure of both MT and neurofilament (NFL) networks as well as swelling of the nerve endings were demonstrated. These changes in the level of key cytoskeletal proteins in the brain of the autistic-like TSC mice suggest the possible molecular mechanisms responsible for neuroplasticity alterations in the ASD brain.

## 1. Introduction

Tuberous sclerosis complex (TSC) is a rare single-gene multisystem disorder with an incidence of 1:6000, with manifestations that are characterized by hamartomas formation that can affect almost every organ, including the brain [1,2,3]. TSC is caused by heterozygous mutations in the *TSC1* (chromosome 9q34) or *TSC2* (16p13.3) genes, encoding for Hamartin and Tuberin proteins, respectively [4,5]. About 70% of individuals with TSC have a *TSC2* mutation, which evokes more severe neurological manifestations, including intellectual disability, than mutations in the *TSC1* gene [4]. Mutations in either *TSC1* or *TSC2* result in dysfunction of the TSC heterotrimeric protein complex (TSC1-TSC2-TBC1D7), which ceases to act as a GTPase-activating protein complex for the small G-protein Rheb, leading to Rheb disinhibition and ultimately causing a constitutive widespread overactivation of the mechanistic/mammalian target of rapamycin complex 1 (mTORC1) [6]. In the central nervous system (CNS), both *TSC1* and *TSC2* are expressed during childhood and adulthood, playing a fundamental role in the regulation of myelination, dendritic spine formation, and axon guidance, as well as dendritic arborization, promoting normal synaptic formation and function [7,8,9,10]. Loss of *TSC1* or *TSC2* leads to perturbed spine structure and density and impaired axon guidance [11,12]. In turn, the mTORC1 broadly regulates cell homeostasis and is involved in a wide spectrum of major cellular processes, including cell growth, proliferation, local protein synthesis, synaptogenesis, synaptic pruning, and autophagy [7,13,14]. As a major regulatory protein, mTORC1 controls dendritic growth and maturation [15]. TORC1-mediated translational control of growth cones plays a key role in axon guidance during development [15]. Taking into account the complexity of mTOR-regulated processes, even subtle alterations in the mTOR-dependent signalling cascade may lead to severe neurological defects typical for both neurodevelopmental, such as autism, and neurodegenerative disorders [16,17,18].

About 90% of individuals (children) with TSC present diverse CNS manifestations [5,19]. Defects in the mTOR pathway are frequently associated with brain malformations and tumours, including epilepsy, mental retardation, intellectual disability/cognitive deficits, anxiety, attention-deficit hyperactivity disorder, and sleep disorders, as well as autism spectrum disorders (ASD) [3,20,21]. Accumulating evidence suggests that abnormal signalling in the mTOR-dependent pathway may serve a critical role in the pathogenesis of ASD in patients with TSC [2,22,23,24,25]. Autism with an increased degree of cognitive impairment has been reported as being much more frequent in TSC individuals than in the general population [20]. Recent studies demonstrated that TSC is associated with ASD in up to 50% of individuals and, in turn, accounts for 1–4% of overall autism cases (although with large variation depending on factors such as country and diagnostic criteria) [1]. About 50–60% of the children with TSC meet the diagnostic criteria for ASD, and the pattern of ASD symptoms diagnosed in TSC individuals resembles that observed in non-syndrome cases [6,13,26,27]. Moreover, it was shown that inhibition of mTOR improves ASD-like behaviours related to Tsc2 or Tsc1 haploinsufficiency (in several *Tsc1*^+/−^ or *Tsc2*^+/−^ animal models), pointing to mTOR attenuation as an effective strategy in various mTOR-related brain dysfunctions, including autism [1,2,25,28,29,30,31].

Given the strong association with ASD and the capacity to diagnose TSC before the onset of atypical development, TSC has been considered an ideal model to prospectively investigate the emergence of ASD within the first years of life [27]. Although individuals with TSC are at high risk of developing ASD, the relationship between these conditions as well as the molecular mechanisms underlying this phenomenon are still poorly understood.

A growing body of evidence indicates that the neurobiological basis of both ASD and TSC-related neurological symptoms may be due to abnormal synaptic transmission, plasticity, and alterations in neuronal connectivity as a result of pathology within synapses [21,31,32]. On the other hand, alterations in neural network connectivity in patients with TSC may result from defects in RHOA-mediated regulation of cytoskeletal dynamics during neuronal development [33]. By combining cytological and immunohistochemical analyses of tubers from TSC patients, Ferrer et al. have shown that alteration of microtubules (MTs) biology could contribute to TSC neuropathology [34]. Additionally, in a study conducted by Jiang et al., *Tsc1*- and *Tsc2*-null cells exhibited abnormal MTs arrangement in the subcortical region, highlighting the role of the TSC2/mTOR in regulating both the structure and function of MTs [35]. Unfortunately, the precise molecular alterations and mechanisms remain largely unknown. A growing body of research, including our previous reports, has provided evidence that abnormal brain development, underlying the aetiology of ASD, may be a result of various abnormalities in neuronal cytoskeletal elements, microtubule-associated proteins (MAPs), and synaptic molecules that together affect wide aspects of neuronal communication [36,37,38]. Thus, the common pathway in both autistic and TSC brain development could be a perturbation of neuronal connectivity in which information-processing abnormalities could be associated with mTOR-related pathological alterations in the morphology of neurons, anomalies in synapses, as well as destabilization of the neuronal cytoskeletal network.

Both the formation and development of the CNS, along with a suitable brain wiring/connectivity, is an extremely complex process, governed by the communication and careful coordination of the neuronal cytoskeleton [39]. In order to facilitate the correct functioning of neuronal circuits during the transmission of electrical and chemical signals along and between neurons, both a dynamic and stable cytoskeleton is required [40]. The neuronal cytoskeleton comprises three main families of filaments: actin-based microfilaments (MFs) (diameter of 8 nm), intermediate filaments (IFs) (diameter ranging from 7–11 nm), and microtubules (MTs) (diameter of 25 nm); forming a complex network capable of processing intracellular information [41,42].

The MT cytoskeleton is fundamental in neuronal development, since it determines the maintenance of the correct structure of neurons and their synapses along with the proper course of neurotransmission [43,44]. This is highlighted by the wide range of nervous system abnormalities linked to pathologically altered MT-mediated processes. The destabilization of the MT system, especially anomalies in the MAP-Tau protein, has been revealed in neurodegenerative diseases, in particular in Alzheimer’s disease (AD) and other tauopathies [45,46,47,48,49,50,51]. Moreover, there is sufficient evidence to suggest the implications of defective/deregulated mTOR signalling in MAP-Tau pathology observed in these neurodegenerative disorders [52,53,54,55,56,57,58].

A recent study suggests the involvement of a dysfunctional MT network also in the aetiology of neurodevelopmental disorders, including ASD [59,60,61,62,63,64,65,66]. In addition, novel and most interesting data, together with our previous reports, indicate the potential involvement of MAP-Tau abnormalities in the molecular mechanisms underlying the pathogenesis of several neurodevelopmental disorders, in particular, those featuring hyperactivation of the mTOR pathway, including ASD [37,67,68,69,70]. Nevertheless, it is unknown whether anomalies in MAP-Tau actively contribute to the pathology of these diseases, as was originally proposed by Tai and collaborators (in which 50% in Tau reduction was sufficient to prevent or diminish autism-like behaviour in animal models of ASD), or whether they are rather the result of the neurotoxic effects of mTOR overstimulation [67,68]. Moreover, studies linking the direct effect of mTOR signalling deregulation on the cytoskeletal proteins dyshomeostasis, especially on the fundamental MTs building blocks, such as α/β-tubulin, MAP-Tau, and other related MAPs, which have been suggested in the pathology of neurodevelopmental syndromes, including autism, are still relatively sparse or lacking. Therefore, the aim of our study was to determine the effect of haploinsufficiency of Tsc2, causing mTOR pathway (mTORC1) hyperactivation on the expression of key structural MT-associated proteins. Specifically, we determined the effect of genetically up-regulating mTOR signalling on disturbances in proteostasis of α/β-tubulin, MAP-Tau, MAP1B, MAP2A/B and MAP2C/D, MAP6 (STOP), and NF-L; key proteins of the neuronal and MT cytoskeleton responsible for the correct neurotransmission and brain connectivity. In 2-month-old TSC (*Tsc2*^+/−^) and wild-type (*Tsc2*^+/+^) mice, we analysed the hippocampus, cerebral cortex, and cerebellum, the brain structures that control many of the executive functions of the brain, including higher-order cognitive processes.

Our study indicated brain-structure-dependent abnormalities in MAP-Tau protein, as well as a significant decrease in both MAP1B and NF-L proteins levels, following mTOR overstimulation. Along with the observed molecular changes, pathological irregularities in the ultrastructure of both the MTs and NFLs skeleton were detected. In summary, all these findings provide new evidence suggesting destabilization and thus dysfunction of the neuronal cytoskeleton network as a result of mTOR signalling overstimulation. Our results emphasize the importance of excessive mTOR activation as a possible trigger of molecular cascade leading to neuronal cytoskeleton protein dyshomeostasis.

## 2. Results

### 2.1. Tsc2 Haploinsufficiency Is Associated with Alterations in the Cytoskeleton Network Organization

The cytoskeleton of neurons not only affects the maintenance of neuronal cell shape and intracellular trafficking of presynaptic and postsynaptic components, but it can also act as a remote control for synaptic strength. It effectively supplies the synapse in neurotransmitter receptors; it also establishes and maintains neural circuitry, crucial for synaptic signalling and cognitive functions. Therefore, we performed the ultrastructure analysis of the cytoskeleton in the brains of *Tsc2*^+/−^ mice using transmission electron microscopy (TEM). Because of the nano-size of cytoskeletal fibres, TEM is a fundamental tool and a gold standard to investigate the ultrastructure of the cytoskeleton.

Selected electronograms showing the control brain tissue from *Tsc2*^+/+^ mice and some alterations in cytoskeleton ultrastructure in *Tsc2*^+/−^ mice are presented in Figure 1—Panel I.

TEM analysis of neuronal cytoskeleton networks showed unchanged and proper organization of its components including microtubules (MTs) and neurofilaments (NFLs) in the hippocampus (CA1/CA2), cerebral cortex, and cerebellum of control *Tsc2*^+/+^ mice (Figure 1—Panel I-a, c, e). The cytoplasm of neurons had a normal appearance and exhibited a proper distribution of MTs. We observed bundles of mostly straight, long, and tightly packed filaments (MTs) in the hippocampus (CA1/CA2), cerebral cortex, and cerebellum of control *Tsc2*^+/+^ mice (Figure 1—Panel II-a; Figure 1—Panel III-a,b; Figure 1—Panel IV-a). Additionally, in Figure 1—Panel II-a and in Figure 1—Panel III-b, we also showed the MTs in cross-section. MTs appear to remain at regular intervals from each other. Moreover, the content of neurofilaments (NFLs) (intermediate filaments) was unchanged in all analysed brain regions in control animals (Figure 1—Panel III-a, b; the yellow arrows point to adjacent MTs and the brown arrows to NFLs for comparison). The nerve endings did not show significant changes in morphology in all examined brain structures in *Tsc2*^+/+^ control mice (Figure 1—Panel I-a,c,e; Figure 1—Panel II-a; Figure 1—Panel III-a,b; Figure 1—Panel IV-a).

In contrast, the electronograms of selected brain regions such as the hippocampus (CA1/CA2), cerebral cortex, and cerebellum of *Tsc2*^+/−^ mice revealed some pathological changes in the ultrastructure of cytoskeletal elements. We observed reorganization or fragmentation of the MTs networks (arrows in Figure 1—Panel II-b–d, f; Figure 1—Panel III-c–f; Figure 1—Panel IV-b–i). Furthermore, microtubules in mice with Tsc2 haploinsufficiency were sparsely packed in neural endings, fragmented and significantly shorter in all examined brain structures compared to control animals (Figure 1—Panel II-b,d,e–g; Figure 1—Panel III-c–g; Figure 1—Panel IV-b–i). In addition, the nerve endings revealed features of swelling (Figure 1—Panel II-b–d; Figure 1—Panel III-c–f; Figure 1—Panel IV-b–f).

### 2.2. Decreased NF-L Protein Is a Key Feature of the Cerebral Cortex from Tsc2^+/−^ Mice

To evaluate the expression of essential structural proteins responsible for the formation of the MTs and NFLs, we measured the level of α/β-tubulin and NF-L protein as the fundamental MTs building blocks and as the most abundant neurofilament protein in axons, respectively. Our analysis revealed that the level of α/β-tubulin was not changed in any of the analysed brain structures (Figure 2A,C). In turn, the analysis of the immunoreactivity of NF-L protein showed a significant decrease (by about 20%, *p* = 0.0445) in the protein level of NF-L only in the brain cerebral cortex of *Tsc2*^+/−^ mice, compared to the control group (Figure 2B,C).

### 2.3. Tsc2^+/−^ Mice Exhibit Increased MAP-Tau Phosphorylation in a Brain-Structure-Dependent Manner

One of the key factors that regulate MTs stability is MAP-Tau protein. Its excessive phosphorylation decreases its MT-binding capacity, leading to MTs disruption. Thus, in the next step, we analysed the protein level of Tau and its phosphorylation status at three different specific residues: (Ser396), (Ser416), and (Ser199/202). The effect of Tsc2 haploinsufficiency on the Tau level and phosphorylation of MAP-Tau were investigated in the hippocampus, cerebral cortex, and cerebellum by using Western blotting. Our study indicated a significant increase in the protein level of total Tau in the hippocampus (by about 17%, *p* = 0.0348) and cerebral cortex (by about 26%, *p* = 0.0240) of *Tsc2*^+/−^ mice (Figure 3A,E). In turn, the level of Tau in the cerebellum was significantly reduced by about 25% (*p* = 0.0413) in *Tsc2*^+/−^ mice compared to the control (Figure 3A,E). The analysis of Tau protein phosphorylation revealed a significantly increased level of p-Tau, phosphorylated at (Ser396) (by about 38%, *p* = 0.0086) in the cerebral cortex, and (by about 29%, *p* = 0.0350) in the cerebellum of *Tsc2*^+/−^ mice, compared to the respective control groups (Figure 3B,E). In the hippocampus, Tsc2 haploinsufficiency has no effect on the immunoreactivity of p-Tau (Ser396) (Figure 3B,E). In turn, the level of Tau phosphorylated at (Ser416) in the neurons of the hippocampus and cerebral cortex of *Tsc2*^+/−^ mice was unchanged versus the respective control groups (Figure 3C,E). However, in the cerebellum, we revealed a significant (*p* = 0.0358) increase by about 54% in the level of Tau phosphorylation at (Ser416) in *Tsc2*^+/−^ mice, compared to control animals. Additionally, our study showed a significant increase in the immunoreactivity of phospho-Tau, phosphorylated at (Ser199/202) in the neurons of the hippocampus (by about 38%, *p* = 0.0461) and cerebral cortex (by about 42%, *p* = 0.0196) of *Tsc2*^+/−^ mice, compared to respective controls (Figure 3D,E).

### 2.4. Tsc2 Haploinsufficiency Is Associated with Decreased MAP1B in the Hippocampus and Cerebral Cortex

Given the importance of other proteins responsible for maintaining the integrity of the cytoskeleton, in the next step we analysed the expression of MAP proteins that affect the assembly and stability of the MT networks and thus condition the maintenance of a proper synaptic transmission, such as MAP1B, MAP2A/B, MAP2C/D, and MAP6 (STOP). Our analysis showed the effect of Tsc2 haploinsufficiency on the immunoreactivity of MAP1B both in the hippocampus and in the cerebral cortex, which was significantly reduced by about 18% in the hippocampus (*p* = 0.0419), and by about 25% (*p* = 0.0109) in the cerebral cortex, compared to the respective control groups (wild-type animals) (Figure 4A,E). The levels of the other examined MAPs were not changed in any of the analysed brain structures (Figure 4B–E).

## 3. Discussion

The neuronal cytoskeleton is one of the proposed players with an enabling role in the pathogenesis of both ASD and TSC. Dysregulation of the MT component of the cytoskeletal network during early brain development is considered to be a common insult for the pathogenesis of various neurodevelopmental syndromes, including ASD [39,64,66,71,72,73]. Especially, alterations of the MAP-Tau have been linked to ASD [67,68,70]. Some findings also suggest an enabling role of microtubule pathology in the pathogenesis of TSC. In addition, some clinical and cell studies suggest that the MT organization and cytoskeleton network dynamics could be adversely affected in the TSC brain, which can result in harmful developmental defects [33,34]. Thus, the involvement of the mTOR pathway in the regulation of the neuronal cytoskeleton and MT-dependent intracellular trafficking has been proposed [35,74]. However, to date, no studies have been conducted on the direct impact of dysregulation of mTOR signalling on changes in the levels of cytoskeletal proteins crucial for MT formation and their correct structure, function, and dynamics.

The current study is the first to provide evidence that overstimulation of the mTOR pathway (loss of TSC2 expression) leads to defects in MT assembly and properties of the MT cytoskeleton. We revealed a significant increase in the level of MAP-Tau in the hippocampus and cerebral cortex with a simultaneous decrease in its level in the cerebellum. Additionally, we demonstrated Tau protein hyperphosphorylation at (Ser396), (Ser416), and (Ser199/202) in a brain-structure-dependent manner. Moreover, the significant loss of MAP1B in the hippocampus and cerebral cortex, as well as depletion of NF-L protein level in the cerebral cortex, was detected. Dyshomeostasis of cytoskeletal proteins was accompanied by ultrastructural pathology within the MT and NFL network. Our observations provide insights into how hyperactivity of mTOR may disrupt the MT organization and all cytoskeletal integrity, which may result in the impairment of synaptic structure, function, and plasticity, leading to neurodevelopmental deficits, and increasing the risk of developing ASD in TSC patients.

The mTOR pathway is a key regulator of synaptic protein synthesis; therefore, any disturbances in mTOR signalling have been linked to synaptic and neuroanatomical abnormalities in both syndrome TSC and idiopathic ASD [31]. Overstimulation of mTOR activity induced by mutations in the *TSC* gene, triggers a cascade of aberrant events that translates into the formation of tubers, along with a number of consequences that include anomalies in the synaptic network, altered synaptic transmission and plasticity, and imbalance in the E/I ratio, leading to abnormal development of the neural circuitry [13,20]. This aberrant wiring of neuronal connections formed during development, can increase susceptibility to ASD and lead to cognitive and behavioural deficits. Using resting-state fMRI, electrophysiology, and in silico modelling in Tsc2-haploinsufficient mice, it has been shown that an mTOR-dependent increase in spine density is associated with ASD-like stereotypies and cortico-striatal hyperconnectivity [31]. In turn, post-mortem histological examinations have revealed increased density of both the dendritic spines and excitatory synapses in the brains of ASD individuals [25,75,76]. In addition, our previous findings support and extend the evidence that links mTOR-hyperactivity with synaptic pathology both in Tsc2-haploinsufficient mice [77] and in environmental-triggered rat models of ASD [37,38]. Previously, we revealed that the inactivation of the *Tsc2* gene triggers pathological alterations in both synaptic ultrastructure and in the expression of several key synaptic proteins [77]. However, the interaction between tubers, cognitive dysfunctions, and ASD in TSC individuals during development is still elusive and requires further investigation. One of the suggested convergence points could be the neuronal cytoskeleton.

One of the fundamental structural components of the cytoskeleton, essential for neuronal development and suitable brain connectivity, is the MT network [43,44]. These polarized and extremely dynamic tubes comprising α- and β-tubulin heterodimers that assemble in a head-to-tail fashion, forming 13 linear proto-filaments that wrap around one another, play a pivotal role in a wide spectrum of cellular processes, such as accurate axon guidance and dendrite arborization, neurogenesis, synaptogenesis, myelination, cell proliferation, differentiation, and migration, as well as in polarity of neurons [39,42,44,66,72,78,79]. MT networks are also the essential tracks for long-distance intracellular cargo trafficking (neurotransmitter receptors, synaptic vesicles, cell adhesion molecules, organelles, cell signalling molecules, and mRNAs) along axons and dendrites to establish appropriate signalling pathways; therefore, they provide the proper neuronal connectivity and neurotransmission [39]. The MTs’ structure, function, and dynamics are determined by post-translational α/β-tubulin modifications; their structure is also regulated by interactions with different MT-associated proteins (MAPs) [80,81]. Among many categories of MAPs (motile, motors, nucleators), structural MAPs, especially MAP-Tau, MAP1, MAP2, and MAP6, are thought to play a pivotal role in MT dynamics, stabilization, properties, and functions [72,82,83,84,85].

MAP-Tau, predominantly present in the axonal compartments of neurons, is one of the key proteins involved in the regulation of MT assembly and stabilization, and thus determines proper axonal transport as well as neurite outgrowth and synapse formation [71,86,87,88]. The affinity of Tau to α/β-tubulin, and therefore the ability of Tau to protect MTs against depolymerization and to promote MT assembly, strictly depends on its degree of phosphorylation [89,90,91,92]. Although Tau contains approximately 85 potential phosphorylation sites in its longest isoform, phosphorylation at (Ser396) position seems to play a pivotal role in Tau function and in particular depolymerizes and destabilizes MTs [47,49,93,94,95,96]. Additionally, (Ser199/202) and (Ser416) are critical phosphorylation sites of Tau that have been related to Tau pathology [97,98]. Under pathological conditions, abnormal hyperphosphorylation of Tau (especially its excessive phosphorylation at (Ser396), (Ser199/202), and (Ser416) epitopes) [47,49,93,94,95,96] promotes its detachment from MTs and subsequent aberrant aggregation, missorting to subcellular compartments, leading to synaptic dysfunction and blockade of neuronal signalling [46,99,100,101]. Hyperphosphorylated and aggregated into neurofibrillary tangles (NFTs), Tau is a critical event in several neurodegenerative disorders, collectively known as tauopathies [48,90,102]. There is sufficient evidence to suggest implications of dysregulated mTOR signalling in MAP-Tau pathology observed in these neurodegenerative diseases [52,53,54,55,56]. More recent studies have shown that Tau protein pathology is also involved in several neurodevelopmental disorders, particularly those featuring hyperactivation of the mTOR pathway, including ASD [67,68,103]. Furthermore, our previous studies have revealed the potential involvement of abnormalities in MAP proteins (including MAP-Tau pathology) simultaneously with excessive mTOR activity in the molecular mechanisms underlying the pathogenesis of neurodevelopmental disorders, including ASD [36,37]. In the current study, we revealed a significant increase in the level of Tau protein in the hippocampus and cerebral cortex with a simultaneous decrease in its level in the cerebellum. Moreover, its excessive phosphorylation at (Ser396) in the cortex and cerebellum, at (Ser416) in the cerebellum, and at (Ser199/202) in the hippocampus and cerebral cortex, as a result of increasing mTOR signalling was observed. Our results are consistent with the study by Caccamo et al., who showed that genetically enhanced mTOR activity facilitated Tau pathology by elevating both endogenous Tau levels and phosphorylation in the hippocampi of the TSC2 heterozygous mice [53]. Liu et al. described elevated levels of phosphorylated Tau in the cerebrospinal fluid of patients with TSC, suggesting that overstimulation of the mTOR pathway may be a novel, amyloid-independent tauopathy similar to that seen in AD [52]. Moreover, studies in vitro by Tang et al. have reported that up-regulated mTOR promotes Tau dyshomeostasis by mediating the synthesis, phosphorylation, and aggregation of Tau. The active form of mTOR aberrantly accumulates in NFT-bearing neurons and mediates Tau phosphorylation in AD-related epitopes in vitro, as well as synthesis and aggregation of Tau, resulting in compromised MTs stability [54,55]. The overexpression of mTOR and excessive Tau phosphorylation are considered to be the driving force behind Amyloid β and NFTs, neuropathological hallmarks of AD [56]. Thus, the activation of the mTOR signalling cascade may enhance Tau pathology by elevating the levels of Tau and its abnormal phosphorylation [53,54]. It has been established that mTOR activation increases the translation of Tau mRNA via downstream p70S6 kinase, which could clarify the rise in the level of hippocampal and cortical Tau observed in our experimental conditions. Tau mRNA has 5′ top-like structure that is preferentially regulated by the mTORC1-S6K pathway [97]. Given the evidence linking mTOR to Tau pathology, our findings revealed changes in both Tau protein level and its phosphorylation status in TSC2 heterozygous mice, suggesting a direct link between mTOR signalling and MAP-Tau protein. Our results also highlight the potential importance of mTOR-induced Tau abnormalities in TSC brains as one of the critical processes in the aetiology of various TSC-related neurological symptoms, collectively termed Tuberous Sclerosis Complex Associated Neuropsychiatric Disorders (TAND), in which ASD is highly prevalent. Dysmorphic neurons of tubers from the brains of TSC patients presented cytoskeletal abnormalities that are almost identical to those seen in the neurofibrillary-tangle-containing neurons [104].

Pathological Tau exhibits prion-like activity, as it sequestrates not only normal Tau but also other neuronal MAPs, such as MAP1 and MAP2, and destroys pre-assembled MTs, leading to progressive degeneration of the affected neurons [105,106]. Different MAPs contribute to normal cytoskeleton organization and dendritic arborization, which are crucial for the function and formation of neural networks [71,78]. Increasing evidence suggests that the levels of various structural MAPs are altered in neurodevelopmental disorders [72]. However, systematic investigation of the MAP proteins involved in the mediation of autism-like behaviours, as well as the effect of mTOR on these proteins, is lacking.

One of the best-known proteins with microtubule-stabilizing activity is the MT lattice-binding structural MAP1B. MAP1B is essential for the development and function of the nervous system [107]. Expressed in axons, dendrites, and growth cones, this protein participates in the regulation of axonal guidance and elongation, neuronal migration, as well as in the formation and development of dendrites in glutamatergic synapses [84,107,108,109,110]. One study suggests that MAP1B mediates MT stabilization specifically by reducing depolymerization rates [84]. MAP1B can bind not only to MTs but also to actin filaments, establishing MAP1B as a potential link between these two components of the growth cone cytoskeleton [111]. Its deficiency causes abnormal actin microfilament polymerization and altered activity of GTPases that regulate the actin cytoskeleton [112]. The lack of MAP1B immunoreactivity has been associated with impairment in brain development [109,113]. Moreover, the synaptic abnormalities associated with neurotransmission were displayed, such as delayed fusion events of synaptic vesicles, a reduced density of synaptic active zones, decreased synaptic vesicles at presynaptic terminals, along with an increased proportion of excitatory immature symmetrical synaptic contacts in MAP1B-deficient neurons [110]. Liu and colleagues proposed that MAP1B is a potential candidate for intellectual disability and ASD [114]. In addition, genetic studies have shown that MAP1B mutations affect general cognitive ability, causing intellectual disability through a profound whole-brain deficit of white matter with likely disordered or compromised axons [115]. Our study revealed that overstimulation of mTOR evoked a large depletion in MAP1B levels in the hippocampus and cerebral cortex of TSC2 mice, compared to the control. To the best of our knowledge, this is the first study reporting alterations of MAPs in the brains of Tsc2-haploinsufficient mice.

The next member of the structural MAPs family is MAP2, which is restricted to cell bodies and dendrites. Its microtubule-stabilizing activity is associated with its ability to interact with MTs via the α/β-tubulin-binding domain [85,116]. Similarly to Tau, its affinity to bind to MTs is determined by its degree of phosphorylation [116,117]. Excessive MAP2 phosphorylation or calpain-induced degradation lead to the dissociation of MAP2 from α/β-tubulin, thus disrupting the proper structure of MTs [118]. Accumulating evidence suggests a much broader range of MAP2 functions, such as binding to filamentous (F) actin, interaction with the neurofilaments of the cross-bridges between MT and neurofilaments, and recruitment of signalling proteins. The ability of MAP2 to interact with both MTs and F-actin might be critical for neuromorphogenic processes, such as neurite initiation and outgrowth [85]. In several case reports of intellectual disability and neuropsychiatric disorders including ASD, a reduced level of MAP2 has been noted. The lack of MAP2-mediated activity/knockdown of MAP2 demonstrably induced laminar cytoarchitectonic changes which were revealed in adult autistic individuals in a study conducted by Mukaetova-Ladinska et al. [119]. Despite this, the consequences of pathologically dysregulated MAP2 have been sparsely explored. There are also no data on the impact of mTOR pathway impairment on the level of MAP2. In our study, we observed no changes in the level of MAP2 protein.

Besides MTs and actin filaments, neurofilaments are also crucial for the proper organization of the neuronal cytoskeleton and thus the formation of the nervous system. Intermediate filaments, along with neurofilament light polypeptide (NF-L), which is the most abundant IF in axons, are the critical scaffolding components of the axonal skeleton of neurons, which through direct interaction with many synaptic phosphoproteins, support and coordinate the shape of neuronal cells, cytoarchitecture, synaptogenesis, and neurotransmission [120,121]. Neurofilaments along with NF-L constitute the structural core of myelinated axons and modulate the axonal diameter, which is crucial for axonal transport and nerve conduction velocity [121]. Here, we revealed a distinct reduction in the level of NF-L protein. NF-L defects cause multiple neurodegenerative disorders [122,123]. Deficits in neuronal NF-L correlate well with the observed axonal and neuronal atrophy, progressive neurite degeneration, and synaptic disorganization in tissues affected by AD and other progressive, age-related neurological diseases [120]. Mutations in the NF-L gene are responsible for autosomal Charcot–Marie–Tooth disease with different phenotypes [121,124]. Loss of NF-L protein has been demonstrated in several models of traumatic brain injury [125]. NF-L is established as a clinical stroke biomarker [126]. Moreover, neuro-axonal brain damage releases NF-L proteins, which enter the blood. NF-L level is elevated in the serum of ASD individuals and related to symptom severity, suggesting that NF-L may play a role in ASD progression [127]. Elevated NF-L in cerebrospinal fluid was exhibited also in the TSC cohort in the study by Liu et al. [128]. Adults with TSC showed phenotypic overlap with frontotemporal dementia. These results support a possible clinical continuum between TSC-associated neuropsychiatric disorders and neurodegenerative illness and highlight mTOR as a convergence point/link between neurodevelopmental and neurodegenerative processes, with a potential role of cytoskeleton dysfunction.

In the current research, we identified alterations in cytoskeleton proteostasis in combination with cytoskeleton ultrastructure impairments as a result of hyperactivation of mTOR pathway. To summarize, the common pathway in both TSC and autistic brain development could be a perturbation of neuronal connectivity in which abnormalities of information processing could be associated with mTOR-related pathological alterations in the neuronal MT network. The results of the current study support the hypothesis that MAPs abnormalities in *Tsc2^+/−^* mice are the result of the neurotoxic effects of mTOR up-stimulation rather than the reason that actively contributes to the pathology of TSC-related disorders. This is indirectly confirmed by the literature data. Inhibiting the activity of mTORC1 with rapamycin significantly enhanced the stability of MTs through resisting depolymerization in HeLa cells [74]. Furthermore, Laks et al., provide evidence that chronic rapamycin induces MT stability in a MAP1B-dependent manner in glioblastoma cells [129]. Inhibition of mTOR by rapamycin significantly affected MT assembly, elongation, and stability in a study by Choi et al., revealing a new role for mTOR signalling in the regulation of MT structure and function, which appears to be mediated by MAP protein-Bik1p [130]. In turn, Jang et al. found that destabilizing MTs increases mTORC1 activity in *Drosophila*, suggesting the new role of MTs in mTOR signalling [131].

Summarizing, our study indicated pathological alterations in the cytoskeleton network organization together with dyshomeostasis in various MAP proteins as a result of loss of Tsc2 expression, pointing to a novel function of the mTORC1 in microtubule network organization. We suggest the potential involvement of pathologically changed neuronal skeleton network as one of the molecular mechanisms underlying the neuropathology of ASD in TSC individuals. However, at this stage of research, we have no evidence for this, which is the most serious limitation of the present study. Cytoskeletal proteins such as MAPs play a pivotal role in synaptic transmission and postsynaptic plasticity. Therefore, any pathological alterations in the organization and function of MTs may lead to defective neuronal communication, which is a fundamental causative factor and key mechanism underlying ASD pathology. This indicates the need for further studies using specific stabilizing compounds of the MT cytoskeleton in order to demonstrate direct connection between cytoskeletal dysfunction and autism in TSC patients. It would be worth investigating whether specific cytoskeletal modulators could improve autistic-like behaviour in Tsc2-haploinsufficient animals.

## 4. Materials and Methods

### 4.1. Animals

Transgenic mouse models of TSC recapitulate core features of ASD and are suitable models to examine the contribution of aberrant neuronal connectivity to ASD.

Young male B6;129S4-Tsc2^tm1Djk^/J mice (*Mus musculus*) were provided by The Jackson Laboratory and bred in The Laboratory For Genetically Modified Animals of the Mossakowski Medical Research Institute PAS (Warsaw, Poland), which breeds small rodents according to the SPF standard. Animals were analysed at postnatal days 60 (PND60). The animals were maintained under controlled temperature and humidity conditions with a 12 h light/dark cycle. All of the experiments conducted on animals were approved by the Local Ethics Committee for Animal Experimentation in Warsaw (reference numbers WAW2/081/2019, WAW2/142/2019, and WAW2/045/2020) and were carried out following the EU Directive 2010/63/EU for animal experiments. Every effort was made to minimize the number of animals used and reduce the amount of pain and distress. To avoid the litter effect, for every test/assay, data from at least three different litters were analysed. To determine the genotype, DNA was isolated from the tail and analysed as previously described [77].

### 4.2. Transmission Electron Microscopy (TEM) Analysis

TEM analysis was performed as described previously [77]. Ten animals (5 each in control and experimental groups) at PND 60 were anaesthetized with ketamine and xylazine (100 and 10 mg/kg, respectively, i.p.) and perfused through the ascending aorta initially with 0.9% NaCl in 0.01 M sodium-potassium phosphate buffer, pH 7.4, and afterward with a mixture of 2% paraformaldehyde and 2.5% glutaraldehyde in 0.1 M cacodylate buffer, pH 7.4, at 4 °C. Small (ca. 1–2 mm^3^) pieces of brain tissue were dissected: (i) from the hippocampus—cornu ammonis (CA) subregions CA1/CA2 and subregions CA2/CA3 (both pyramidal layer and stratum radiatum were collected); (ii) from the cortex—primary somatosensory area, layer 1–4; and (iii) from the cerebellum—Crus2 (both molecular and granular layers). Next, tissue specimens were post-fixed in a mixture of 1% OsO_4_ and 0.8% K_4_[Fe(CN)_6_], dehydrated in a series of ethanol gradients, embedded in epoxy resin, and polymerized at 60 °C for 24 h. Brain tissue blocks were cut into ultrathin sections (60 nm thick), placed on copper grids, and stained with uranyl acetate and lead citrate. Poststaining ultrathin sections on girds were examined using TEM JEM 1011 (JEOL, Tokyo, Japan) operated at 80 kV.

### 4.3. Western Blot Analysis

Immunochemical analysis of protein level and phosphorylation status was per-formed using the Western blotting method in standard conditions. Tissue samples were homogenized, mixed with Laemmli buffer, and denatured at 95 °C for 5 min. After standard SDS-PAGE separation, the proteins were “wet”-transferred to nitrocellulose membranes in standard conditions and used for immunochemical analysis with specific antibodies, followed by chemiluminescent detection. The membranes were washed for 5 min in TBST (Tris-buffered saline with Tween 20 buffer: 100 mM Tris, 140 mM NaCl and 0.1% Tween 20, pH 7.6) and non-specific binding was blocked for 1 h at room temperature (RT) with 2% or 0.5% BSA in TBST or with 5% non-fat milk solution in TBST. Membranes were probed with the following primary antibodies: α/β-tubulin (1:1000), NF-L (1:125), Tau (1:500), pTau (Ser396) (1:250), pTau (Ser199/202) (1:1000), pTau (Ser416) (1:1000), MAP1B (1:500), MAP2 (1:1000), and MAP6 (STOP) (1:250). The membranes were washed three times in TBST, incubated for 60 min at RT with appropriate secondary antibodies (1:8000 anti-rabbit or 1:4000 anti-mouse IgG), and washed again three times in TBST. Antibodies were detected using chemiluminescent reaction and ECL reagent (Amersham Biosciences, Bath, UK) under standard conditions. After each protein detection, the membranes were stripped (25 mM Glycine-HCl, 1% (*w*/*v*) SDS, pH 2; 30 min at room temperature) and re-probed. As first, phosphorylated protein was immunodetected, then the total level of analysed protein, and finally, glyceraldehyde 3-phosphate dehydrogenase (GAPDH) or Vinculin as a loading control. GAPDH (1:50,000) was used as a loading control for low-molecular-weight proteins; Vinculin (1:1000) was used as a loading control for analyses of high-molecular-weight proteins. In all experiments, densitometry analysis of immunoblots was performed using normalization to immunoreactivity of GAPDH or Vinculin. Densitometric analysis and size-marker-based verification were performed with TotalLab software.

### 4.4. Statistical Analysis

The results are expressed as mean values ± S.E.M. In all the analyses, each data-point is from a separate animal. The normality and equality of the group variances were tested using a Shapiro–Wilk test. Differences between the means were analysed using an unpaired Student’s *t*-test for data with normal distributions or a non-parametrical Mann–Whitney U test for data with non-normal distribution. Differences were considered significant at *p* < 0.05. The statistical analyses were performed using GraphPad Prism version 8.0 (GraphPad Software, San Diego, CA, USA).

## Figures and Tables

**Figure 1 ijms-24-07303-f001:**
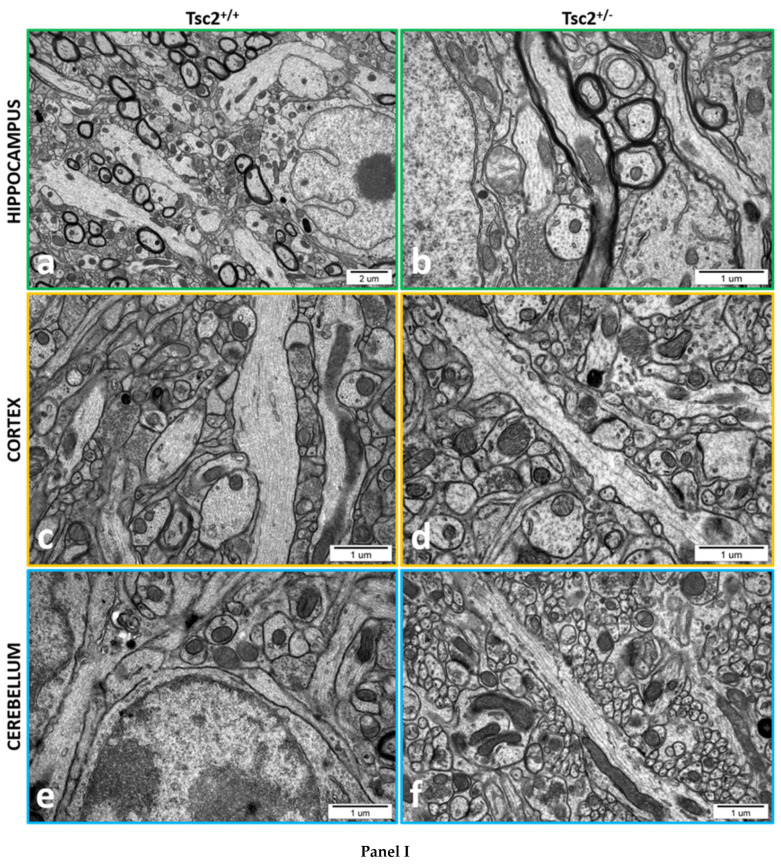
**Panel I.** The effect of Tsc2 haploinsufficiency on the ultrastructure of neuronal cells in the hippocampus (CA1/CA2 region) (**a**,**b**), cerebral cortex (**c**,**d**), and cerebellum (**e**,**f**). Control group (*Tsc2*^+/+^ mice) (**a**,**c**,**e**); *Tsc2*^+/−^ group (**b**,**d**,**f**). Selected electronograms at low magnification are presented. Representative pictures from separate animals for the control (n = 5) and separate animals for the experimental group (n = 5) were presented. **Panel II.** Representative electronograms showing some morphological alterations in the ultrastructure of the neuronal cytoskeleton in the hippocampus. (**a**) Control group (*Tsc2*^+/+^ mice). Ultrastructurally unchanged neuronal cytoskeleton. The normal appearance of the cytoplasm of neurons with unchanged, proper organization of MTs and NFLs. MTs as bundles of mostly straight, long, and tightly packed filaments (MTs, the green arrows; NFLs, the light green arrows). (**b**–**i**) *Tsc2*^+/−^ group. Reorganized or fragmented the MT networks. MTs sparsely packed in neural endings, fragmented, and much shorter (MTs marked with green arrows). Single and short NFLs relative to the controls are visible (NFLs marked with light green arrows). Furthermore, the nerve endings revealed the features of the swelling (nerve endings swelling marked with a white outline). Representative pictures from separate animals for the control (n = 5) and separate animals for the experimental group (n = 5) were presented. **Panel III.** Representative electronograms exhibiting morphological alterations in the ultrastructure of the neuronal cytoskeleton in the brain’s cerebral cortex. (**a**,**b**) Control group (*Tsc2*^+/+^ mice). The normal appearance of the cytoplasm of neurons with unchanged, proper organization of MTs and NFLs. MTs as bundles of mostly straight, long, and tightly packed filaments (MTs marked with yellow arrows; NFLs marked with brown arrows). (**c**–**g**) *Tsc2*^+/−^ group. Reorganized or fragmented the MTs networks. MTs sparsely packed in neural endings, fragmented and much shorter (MTs, the yellow arrows). Single and short NFLs relative to the controls are visible (NFLs, the brown arrows). Swelling of the nerve endings (nerve ending swelling marked with a white outline). Representative pictures from separate animals for the control (n = 5) and separate animals for the experimental group (n = 5) were presented. **Panel IV.** Representative electronograms showing morphological alterations in the ultrastructure of the neuronal cytoskeleton in the cerebellum. (**a**) Control group (*Tsc2*^+/+^ mice). Ultrastructurally unchanged neuronal cytoskeleton. The normal appearance of the cytoplasm of neurons with unchanged, proper organization of MTs. MTs as bundles of mostly straight, long, and tightly packed filaments (MTs, the blue arrows). (**b**–**i**) *Tsc2*^+/−^ group. Reorganized or fragmented MT networks. MTs sparsely packed in neural endings, fragmented and much shorter (MTs, the blue arrows). Single and short NFLs relative to the controls are visible (NFLs, the dark blue arrows). Swelling of the nerve endings (nerve ending swelling marked with a white outline). Representative pictures from separate animals for the control (n = 5) and separate animals for the experimental group (n = 5) were presented.

**Figure 2 ijms-24-07303-f002:**
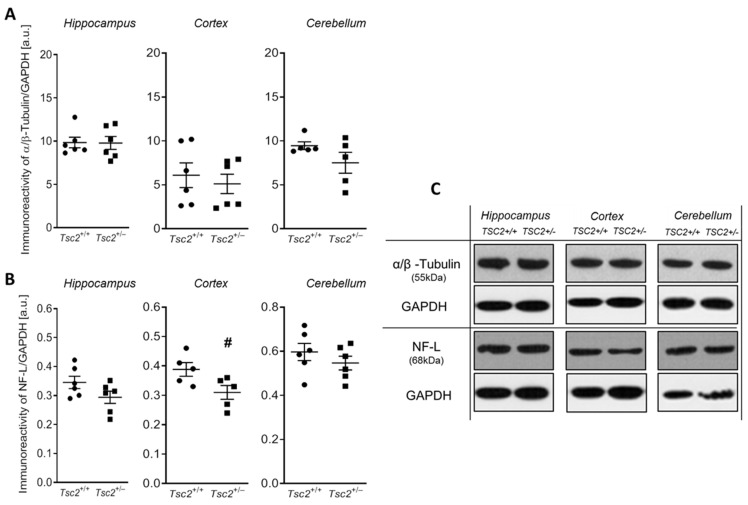
The effect of Tsc2 haploinsufficiency on the protein level of α/β-tubulin and NF-L in the mouse brain. Immunoreactivity of α/β-tubulin and NF-L protein in control and *Tsc2*^+/−^ mice were monitored using Western blot analysis. Densitometric analyses of α/β-tubulin (**A**) and NF-L (**B**) and representative pictures (**C**) are shown. Results were normalized to GAPDH levels. Data represent the mean values ± SEM from n = (5–6) separate animals from three different litters. Data were analysed using Student’s *t*-test. # *p* < 0.05, vs. control (wild-type animals).

**Figure 3 ijms-24-07303-f003:**
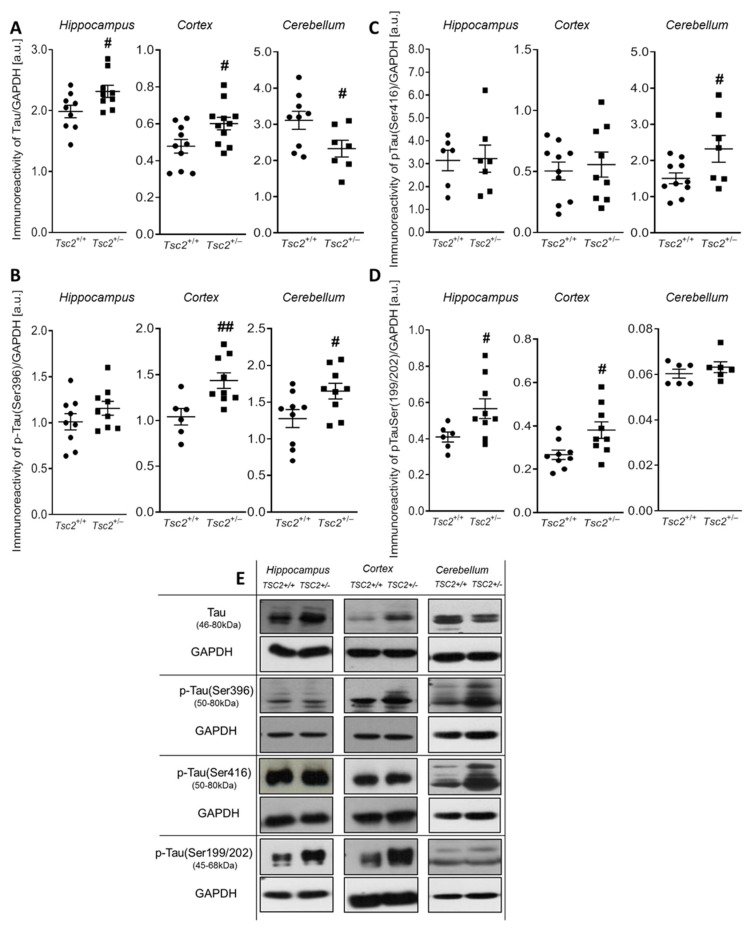
The effect of Tsc2 haploinsufficiency on the protein level and Tau phosphorylation in the mouse brain. Immunoreactivity of total Tau and pTau (Ser396), pTau (Ser199/202), and pTau (Ser416) in control and *Tsc2*^+/−^ mice, was monitored using Western blot analysis. Densitometric analysis of total Tau (**A**), pTau (Ser396) (**B**), pTau (Ser416) (**C**), and pTau (Ser199/202) (**D**), as well as the representative pictures (**E**) in the hippocampus, cerebral cortex, and cerebellum, were shown. Results were normalized to GAPDH levels. Data represent the mean values ± SEM from n = (7–11) separate animals for Tau, n = (6–9) separate animals for pTau (Ser396), n = (6–10) separate animals for pTau (Ser416), and n = (6–9) separate animals for pTau (Ser199/202) from three different litters. Data were analysed using Student’s *t*-test. # *p* < 0.05, ## *p* < 0.01, vs. control (wild-type animals).

**Figure 4 ijms-24-07303-f004:**
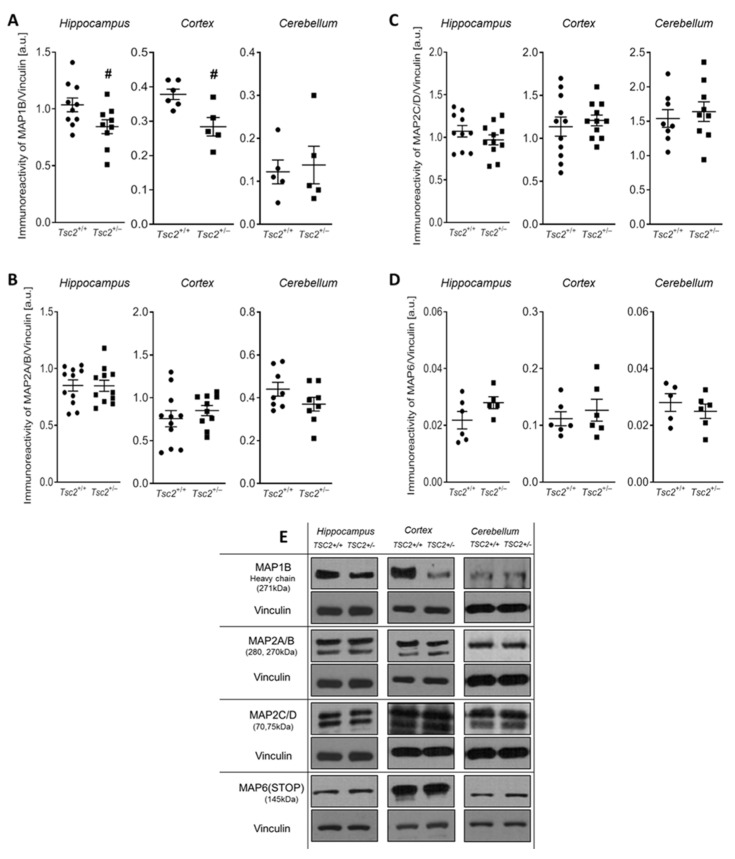
The effect of Tsc2 haploinsufficiency on the protein level of the MAP1B, MAP2A/B, MAP2C/D, and MAP6 (STOP) in the mouse brain. The immunoreactivity of proteins in control and *Tsc2*^+/−^ mice was determined by using Western blot analysis. Densitometric analysis of MAP1B (**A**), MAP2A/B (**B**), MAP2C/D (**C**), and MAP6 (STOP) (**D**) and representative pictures (**E**) in the hippocampus, cerebral cortex, and cerebellum were shown. The results of the densitometric analysis were normalized to Vinculin levels. Data represent the mean values ± SEM from n = (5–10) separate animals for MAP1B, n = (8–11) separate animals for MAP2, and n = (5–6) separate animals for MAP6 (STOP) from three different litters. Data were analysed using Student’s *t*-test. # *p* < 0.05, vs. control (wild-type animals).

## Data Availability

The raw data supporting the conclusions of this article will be made available by the authors, without undue reservation.

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
