# Peer review of "Microtubule Cytoskeletal Network Alterations in a Transgenic Model of Tuberous Sclerosis Complex: Relevance to Autism Spectrum Disorders"

_ijms, 2023, doi:10.3390/ijms24087303_

Round 1

Reviewer 1 Report

The authors aimed to investigate the effect of Tsc haploinsufficiency (known to cause mTOR pathway hyperactivation) on the expression of key structural MT-associated proteins. Towards this goal, they specifically determined the effect of genetically up-regulating mTOR signalling on disturbances in proteostasis of α/β-tubulin, MAP-Tau, MAP1B, MAP2A/B and MAP2C/D, MAP6(STOP), and NF-L, as key proteins of the neuronal and MTs cytoskeleton responsible for the correct neurotransmission and brain connectivity. They utilized brains from 2 month-old TSC (Tsc2+/-) and wild-type (Tsc2+/+) mice to analyse 3 brain regions implicated in executive functions and high-order cognition, namely the hippocampus, cerebral cortex, and cerebellum. Findings from this work indicated brain structure-dependent abnormalities in MAP-Tau protein, as well as a significant decrease in the level of MAP1B and NF-L protein, following mTOR overstimulation. In addition to the molecular changes, thy also observed dysregulation in the ultrastructure of both the MTs and NFLs skeleton. The combination of molecular and ultrastructural evidence points to previously unreported alterations in the neuronal cytoskeleton network due to mTOR signalling overstimulation. This work highlights the the importance of excessive mTOR activation, as a potential molecular trigger of neuronal cytoskeleton protein dyshomeostasis and further supports the potential involvement of pathologically changed neuronal skeleton network in molecular mechanisms underlying the neuropathology of TSC-related neurological symptoms, including ASD.

The role of neuronal cytoskeletal networks in neurological disorders makes this a particularly interesting study. Moreover, the study has a clear rationale, good experimental design and the experiments are well done. It is evident that the authors are experts in the Tsc haploinsufficency model and have previously published papers on the subject. A minor weakness is that the key message of the results is not well articulated in the sub headings of the results section. For instance, 2.1. “The effect of Tsc2 haploinsufficiency on the organization of the cytoskeleton network” is less informative than writing- Tsc2 haploinsufficiency is associated with alterations in the cytoskeleton network organization. I would therefore be inclined to recommend acceptance of the manuscript with minor revisions.  

Minor issues to be addressed:

(i)  In the Results, line 154: 2.1. “The effect of Tsc2 haploinsufficiency on the organization of the cytoskeleton network. ”

This is not a particularly informative subheading for the results therein described.

I would suggest a more informative subheading e.g. Tsc2 haploinsufficiency is associated with alterations in on the cytoskeleton network organization.

(ii)  In the Results, line 258: 2.2. “The effect of Tsc2 haploinsufficiency on the protein level of α/β- tubulin and NF-L.. ”

This is not a particularly informative subheading for the results therein described.

I would suggest a more informative subheading e.g. Decreased NF-L protein is a key feature of the cerebral cortex from Tsc2+/− mice.

(iii)  In the Results, line 275: 2.3. “The effect of Tsc2 haploinsufficiency on the protein level and phosphorylation of MAP-Tau. ”

This is not a particularly informative subheading for the results therein described.

I would suggest a more informative subheading e.g. Cerebral cortex and cerebellum from Tsc2+/− mice exhibit increased MAP-Tau phosphorylation.

(iv)  In the Results, line 309: 2.4 “The effect of Tsc2 haploinsufficiency on the protein level of the other MAP proteins. ”

This is not a particularly informative subheading for the results therein described.

I would suggest a more informative subheading e.g. Tsc2 haploinsufficiency is associated with decreased MAP1B in the cerebral cortex and hippocampus.

Reviewer 2 Report

Congratulations to the authors for this great job

Reviewer 3 Report

Although in principle interesting the claims of the manuscript cannot be rigorously substantiated. In the EM micrographs it is nearly impossible even to a trained eye to see the claimed differences in the structure of the microtubules or NLFs. Higher magnification/resolution images could at least in part resolve this issue. How do we even know that they are microtubules or NLFs without immunological confirmation?

Moreover, if total Tau is elevated in the Hippopampus and the cortex, then why is it that use of the phosphospecific antibodies for the 3 sites selected do not follow the same patterns. In other words is Ser214 hypophosphorylated in the hippocampus?  If it followed the total Tau levels it should be elevated by the same degree as total Tau is in these neurons and as displayed for Ser199/202.  Same for the other two phosphoTau antibodies.

We are not told why these three phosphosites were selected among many more available and the evidence explicitly indicating that when one or all of  these three phosphosites are occupied there is an effect (direct or indirect) on microtubules.

A classical biochemical assay of Tau affinity in extracts/lysates of the three brain tissue types used for exogenously added microtubules would provide critical information towards that end.  

Round 2

Reviewer 3 Report

This version of the manuscript is an improvement from the prior one.

However, all supplemental TEM figures should be main figures. The supplemental figures substantiate much more clearly the claims on cytoskeletal changes because of the higher resolution. I would swap the main TEM with the supplemental or even better I would present both sets as main figures with the extra lower resolution TEMs as supplemental.

Regarding the phsosphoTau/totalTau issue. I do not believe I suggested that the authors should present that calculated ratio. I asked for an explanation as to why they think phosphoTau is lower when total Tau is high. The authors gave a very long winded argument as to why that may be and I believe this argument should be presented in the discussion at least.

However, calculating a ratio is not necessary to argue that when total tau is higher in the mutant, but in the same tissue of the mutant phosphotau (ser214) is less in the mutant than control, then  on average less Tau molecules must be phoshorylated at that site in the mutant tissue. That also needs to be discussed at least as one possible explanation of their data as presented.

Importantly, the authors did not address (to the best of my reading the replies and manuscript), the question of why they chose these 3 phosphosites to assay out of many possible (with available phosphoantibodies). Is there a reason? or it is a random sampling based on available reagents? As I am sure the authors are aware, there are gatekeeper phosphorylations and interactions among phosphorylation sites both facilitating and preventing further Tau phosphorylation.  This is the main reason for asking for this clarification.

Finally, in my opinion the claimed connection of the observed cytoskeletal changes and Autism in Tsc cases remains tenuous and should be argued with more rigor.

Round 3

Reviewer 3 Report

All my concerns and questions were thoroughly addressed. thanks